# Leading causes of death in Asian Indians in the United States (2005–2017)

**Claudia Fernandez Perez**[1,2,3]*, **Kevin Xi**[1,4], **Aditya Simha**[1,5], **Nilay S. Shah**[1,6], **Robert J. Huang**[1,7], **Latha Palaniappan**[1,8], **Sukyung Chung**[1], **Tim Au**[1,9], **Nora Sharp**[1,10], **Nathaniel Islas**[1,11], **Malathi Srinivasan**[1,8]

1 Stanford Center for Asian Health Research and Education (CARE), Stanford University School of Medicine, Stanford, CA, United States of America, 2 Department of Psychology, Neuroscience Program, College of Arts and Sciences, University of Miami, Coral Gables, FL, United States of America, 3 Department of Biochemistry and Molecular Biology, College of Arts and Sciences, University of Miami, Coral Gables, FL, United States of America, 4 Department of Neuroscience, College of Arts and Sciences, Rice University, Houston, TX, United States of America, 5 Department of Biological Sciences, University of California, Davis, Davis, CA, United States of America, 6 Department of Medicine, Division of Cardiology, Northwestern University Feinberg School of Medicine, Chicago, IL, United States of America, 7 Division of Gastroenterology, Stanford University School of Medicine, Stanford, CA, United States of America, 8 Division of Primary Care and Population Health, Stanford University School of Medicine, Stanford, CA, United States of America, 9 Department of Biomedical Engineering, School of Science and Engineering, Tulane University, New Orleans, LA, United States of America, 10 Computational and Systems Biology Interdepartmental Program, University of California Los Angeles, Los Angeles, CA, United States of America, 11 Department of Computer Science, Department of Statistics, California State University East Bay, Hayward, California, United States of America

* cxf507@miami.edu

## Abstract

### Objective

Asian Indians are among the fastest growing United States (US) ethnic subgroups. We characterized mortality trends for leading causes of death among foreign-born and US-born Asian Indians in the US between 2005–2017.

### Study design and setting

Using US standardized death certificate data, we examined leading causes of death in 73,470 Asian Indians and 20,496,189 non-Hispanic whites (NHWs) across age, gender, and nativity. For each cause, we report age-standardized mortality rates (AMR), longitudinal trends, and absolute percent change (APC).

### Results

We found that Asian Indians' leading causes of death were heart disease (28% mortality males; 24% females) and cancer (18% males; 22% females). Foreign-born Asian Indians had higher all-cause AMR compared to US-born (AMR 271 foreign-born, CI 263–280; 175.8 US-born, CI 140–221; p<0.05), while Asian Indian all-cause AMR was lower than that of NHWs (AMR 271 Indian, CI 263–278; 754.4 NHW, CI 753.3–755.5; p<0.05). All-cause AMR increased for foreign-born Asian Indians over time, while decreasing for US-born Asian Indians and NHWs.

**Data Availability Statement:** All relevant data are within the manuscript and its Supporting Information files. Supporting files for all data can be found here: https://www.kaggle.com/

adityasimha/causes-of-death-in-asian-indians-in-usa-data.

**Funding:** Latha Palaniappan R01MD007012 PALO ALTO MEDICAL RESEARCH FOUNDATION https://www.sutterhealth.org/research/pamfri Latha Palaniappan UL1TR003142 STANFORD CENTER FOR CLINICAL & TRANSLATIONAL EDUCATION AND RESEARCH (SPECTRUM) http://med.stanford.edu/spectrum.html The funders had no role in study design, data collection and analysis, decision to publish, or preparation of the manuscript.

**Competing interests:** The authors have declared that no competing interests exist.

## Conclusions

Foreign-born Asian Indians were 2.2 times more likely to die of heart disease and 1.6 times more likely to die of cancer. Asian Indian male AMR was 49% greater than female on average, although AMR was consistently lower for Asian Indians when compared to NHWs.

## 1. Introduction

Asian Indians are one of the fastest growing populations in the United States (US), increasing from 2.3 million individuals in 2005 to 4.8 million in 2020 [1]. Despite this, research in Asian American health often aggregates all Asian subgroups (the six largest in the US are Chinese, Asian Indian, Filipino, Vietnamese, Korean, and Japanese, accounting for >90% of Asian Americans in the US [3]) into a single population, masking differences in health trends between subgroups [2–5]. Understanding population patterns of the prevalent causes of death in specific Asian American subgroups may contextualize biological, sociocultural and lifestyle factors affecting health outcomes in these different populations [6–8], informing targeted interventions and resource allocation to improve population health.

Asian Indians living in the Indian subcontinent and surrounding areas have larger proportions of total death due to cardiovascular-related disease (29%) than Asian Indians living in the US (24%) and other Asian subgroups (28%) [9–12]. Many cultural and biological factors have been proposed [13, 14] to explain this disproportionate burden, including a genetic predisposition to insulin resistance [15] and a "BMI penalty" [16], which describes a higher risk for disease at lower BMI. Unlike the Hispanic community, in which population growth due to immigration has tapered [17], US South Asian (including Asian Indian) population growth has been driven by immigration since the 1960s, most recently with migrant workers in the technology industry, who wish to stay in the US long-term [18, 19]. Blending of American and Asian Indian cultures may impact modifiable health risks [20, 21], supporting place of birth as an important determinant of health. Heart disease mortality declines over the past two decades were less significant for Asian Americans than for non-Hispanic Whites (NHW), but heart disease mortality rates actually increased in Asian Indians in the early 2000s [3, 9]. Unfortunately, national surveillance inadequately samples Asian American subgroups [7], preventing adequate characterization of their longitudinal health trends.

Using mortality data from the National Center for Health Statistics (NCHS) [22] from 2005–2017, we characterized trends in all-cause and cause-specific mortality rates from the top 10 causes of death in Asian Indians in the US, examined overall, by age, and by place of birth.

## 2. Methods

### 2.1 Study data

This study is considered not human subject research by the Stanford Institutional Review Board (protocol number 53429). We examined US mortality records from the National Center of Health Statistics (Hyattsville, MD), containing the Centers for Disease Control and Prevention database of death certificates from 2005–2017, under a data use agreement, which was fully anonymized prior to analysis. This was a retrospective study using fully de-identified data, and it was approved by the Stanford IRB. The US standard certificate of death contains detailed demographic information for each decedent, including race/ethnicity, age, sex (male or female), place of birth, and immediate and underlying causes of death (listed as primary, secondary, and tertiary causes). We used the primary cause as the cause of death in our study.

We characterized the underlying cause of death using the following International Classification of Diseases, 10th revision (ICD-10) codes and subcodes: heart diseases (I00-I09, I20-51), malignant neoplasms (C00-C97), heart failure (I50), hypertensive diseases (I11, I13), chronic lower respiratory diseases (J40-J47), accidents (unintentional injuries) (V00-V99, W00-W99, X00-59, Y85-Y86), cerebrovascular diseases (I60-I69), Alzheimer's disease (G30), diabetes mellitus (E10-E14), influenza and pneumonia (J09-J18), chronic liver diseases (K70-K77), and nephritis and nephrosis (N00-N08). These categories were chosen from causes highlighted in previous studies, and account for 85.6% of all Asian Indian deaths [9]. Decedents categorized as more than one ethnicity or as "Other Asian", and those missing relevant information pertaining to cause of death, year of death, age, or race/ethnicity, were excluded.

We compared mortality between foreign- and US-born Asian Indian males and females with NHW mortality in the same categories. The decedents' age at death was grouped in 5-year intervals, from ages 1–79. Special age brackets were created for the 0–1 age group, the 1–19 age groups, the 70–79 age group, and the 80–99 age group, in order for census data to be comparable to survey data [10].

Linear interpolation of 2000 and 2010 US Census data was used to calculate population sizes [10, 17]. Annual population was estimated using the 2005–2017 1-year American Community Survey (ACS) data [1], which stratifies the population by nativity, race (including Asian subgroup details), age bracket, and sex (male or female). ACS data was used to accurately estimate foreign-born and US-born Asian Indian populations by applying the nativity percentages to each census projection. The 2010 US Census population was the reference used for age-standardization.

The 2003 US standard death certificate disaggregated Asian race into six Asian subgroups: Chinese, Asian Indian, Filipino, Vietnamese, Korean, and Japanese, as well as other Asian. States adopted the 2003 standard certificate revision on a rolling basis [23]. Annualized data accounted for this rolling adoption by including the population of a state in the overall population denominator starting with the year in which the state adopted the 2003 US standard death certificate. Accordingly, Asian Indian decedents were included in analysis only when annual state-level data were available for both decedent frequency (death certificates) and population size (US Census and ACS data).

## 2.2 Statistical analysis

We calculated age-standardized mortality rate (AMR) as deaths per 100,000 person-years, for each ICD-10 coded cause of death from 2005–2017. We then sorted ICD-10 coded data by NCHS category [2], aggregating causes of death into 14 similar groups (e.g. "Diseases of heart", "Malignant neoplasms"). The top 10 causes of death by overall mortality rate were further analyzed to understand the greatest burden of disease in the Asian Indian population. The AMR was calculated and stratified by place of birth (foreign-born vs. US-born) and sex (male vs. female) for each cause of death. All-cause mortality was also examined as the sum of all AMRs from any cause stratified by race/ethnicity, place of birth, and sex. To examine differences in mortality trends between Asian Indian subgroups and NHWs, we used linear regression to estimate the trendline for all-cause mortality, as well as the top 10 specific causes from 2005–2017 by racial/ethnic group, place of birth, and cause of death. Analyses were conducted in R version 4.02, using the epitools packet of statistical tools for direct age-standardization and comparison of standardized rates, and using a 95% confidence interval. Results were determined to be significant in the case of non-overlapping confidence intervals ($p < 0.05$).

## 3. Results

A total of 73,470 (68,100 foreign-born & 5,370 US-born) Asian Indian deaths were identified in the US between 2005–2017 (**Table 1**) [24]. Overall, the leading causes of death amongst Asian Indians in this time period, in descending order by AMR, were: heart diseases (AMR 75.6 per 100,000, CI[74.5–76.7]), malignant neoplasms (58.6, CI[57.7–59.6]), accidents (19.6, CI[19.1–20.2]), diabetes mellitus (17.5, CI[17.0–18.1]), cerebrovascular diseases (16.6, CI[16.1–17.1]), influenza and pneumonia (16.0, CI[15.5–16.5]), Alzheimer's disease (11.6, CI[11.2–12.0]), chronic liver diseases (10.4, CI[10.0–10.8]), nephritis and nephrosis (7.2, CI[6.9–7.6]), and chronic lower respiratory diseases (6.1, CI[5.7–6.4]) (**Table 2**).

### 3.1 All-cause mortality

All-cause mortality rates were higher in males (AMR 323.6 per 100,000 foreign-born, CI[306–346] vs. 241.18 per 100,000 US-born, CI[172–339]) compared to females (AMR 227.5 in foreign-born, CI[217–241] vs. 145.6 in US-born, CI[99.0–213]), an average of 48.8% difference in AMR. US-born Asian Indian females had lower all-cause mortality across a majority of study years compared to their foreign-born counterparts, while this difference was not significant in men (**Fig 1**). All-cause mortality differences were largely due to differences in heart disease (38% of difference) and cancer mortality rates (21% of difference) between US-born and foreign-born populations. Asian Indian AMR was lower compared to NHWs for cause-specific and all-cause mortality, regardless of nativity (**Fig 1**).

**3.1.1 Trends in all cause-specific mortality rates.** Foreign-born Asian Indian all-cause mortality rates consistently trend in the opposite direction of both US-born Asian Indians and NHWs. All-cause mortality rates trended downwards for NHWs (-1.2% per year in males, -1.0% per year in females) and US-born Asian Indians (-2.7% per year in males, -2.0% per year in females), while rising in foreign-born Asian Indians (+0.8% per year in males, +1.0% per year in females). In terms of absolute AMR difference from 2005 to 2017, all-cause mortality

**Table 1. Characteristics of Asian Indian and non-Hispanic White decedents in the United States, 2005–2017.**

| Population Statistics | Asian Indian | | | Asian Indian US-Born | | Asian Indian Foreign-born | | Non-Hispanic White | | |
|---|---|---|---|---|---|---|---|---|---|---|
| | **All** | **Male** | **Female** | **Male** | **Female** | **Male** | **Female** | **All** | **Male** | **Female** |
| Population Size | 2,493,572 | 1,292,768 | 1,200,804 | 339,501 | 322,225 | 953,267 | 878,579 | 229,456,801 | 113,221,153 | 116,235,648 |
| % of Total Demographic Population | 100% | 52% | 48% | 14% | 13% | 38% | 35% | 100% | 49% | 51% |
| **Age demographics (%)** | | | | | | | | | | |
| 0–19 | 26.0% | 26.0% | 26.0% | 74.0% | 75.0% | 9.0% | 8.0% | 26.0% | 27.0% | 25.0% |
| 20–39 | 39.0% | 39.0% | 40.0% | 22.0% | 20.0% | 46.0% | 47.0% | 25.0% | 26.0% | 25.0% |
| 40–59 | 23.0% | 23.0% | 22.0% | 3.1% | 4.2% | 30.0% | 29.0% | 28.0% | 29.0% | 28.0% |
| 60+ | 12.0% | 12.0% | 12.0% | 0.9% | 0.8% | 16.0% | 16.0% | 20.0% | 18.0% | 22.0% |
| **Mortality Statistics** | Asian Indian | | | Asian Indian US-Born | | Asian Indian Foreign-born | | Non-Hispanic White | | |
| | All | Male | Female | Male | Female | Male | Female | All | Male | Female |
| Total # Deaths | 73,470 | 43,333 | 30,137 | 3,271 | 2,099 | 40,062 | 28,038 | 1,971,663 | 972,795 | 998,868 |
| % of Total Deaths | 100% | 59% | 41% | 4.0% | 3.0% | 55% | 38% | 100% | 49% | 51% |
| **Age demographics (%)** | | | | | | | | | | |
| 0–19 | 2.0% | 2.1% | 1.8% | 30.0% | 35.0% | 0.6% | 0.4% | 0.6% | 0.7% | 0.4% |
| 20–39 | 6.6% | 7.8% | 4.7% | 40.0% | 25.0% | 6.1% | 3.8% | 3.5% | 4.7% | 2.3% |
| 40–59 | 17.0% | 20.0% | 14.0% | 11.0% | 12.0% | 20.0% | 14.0% | 12.0% | 14.0% | 10.0% |
| 60+ | 81.0% | 78.0% | 85.0% | 21.0% | 30.0% | 81.0% | 87.0% | 89.0% | 87.0% | 92.0% |

**Table 2.  Age-standardized mortality rates from leading causes of death in Asian Indians in the United States by sex and nativity, 2005–2017.**

| AMR 2005–2017 | Overall Asian Indian | | | Asian Indian Female | | | Asian Indian Male | | | Overall Non-Hispanic White | | |
|---|---|---|---|---|---|---|---|---|---|---|---|---|
| **Cause of Death** | All | Foreign-born | US-born | All female | Foreign-born | US-born | All male | Foreign-born | US-born | All | Female | Male |
| Heart disease | 75.6 (74.5–76.7) | 76.9 (75.8–78.1) | 35.2 (30.9–40.1) | 55.6 (54.3–57) | 57.0 (55.6–58.6) | 20.8 (16.3–26.3) | 94.5 (92.8–96.3) | 95.8 (94–97.6) | 50.2 (42.8–58.7) | 162.9 (162.8–163) | 129.0 (128.9–129.2) | 204.0 (203.8–204.2) |
| Malignant neoplasms | 58.6 (57.7–59.6) | 59.3 (58.3–60.4) | 37.2 (32.6–42.3) | 56.2 (54.8–57.6) | 57.0 (55.5–58.5) | 34.9 (28.8–42) | 61.4 (60.1–62.9) | 62.0 (60.6–63.5) | 39.8 (33–47.8) | 167.9 (167.7–168) | 143.5 (143.3–143.6) | 200.9 (200.7–201.2) |
| Accidents (unintentional injuries) | 19.6 (19.1–20.2) | 20.2 (19.5–20.9) | 17.8 (15.7–20.4) | 11.1 (10.6–11.7) | 11.4 (10.7–12.2) | 7.8 (5.8–10.5) | 27.5 (26.7–28.4) | 28.3 (27.2–29.4) | 27.6 (23.8–32.2) | 59.4 (59.3–59.5) | 37.5 (37.4–37.6) | 82.5 (82.4–82.7) |
| Diabetes mellitus | 17.5 (17–18.1) | 17.7 (17.2–18.3) | 9.7 (7.4–12.5) | 14.3 (13.6–15) | 14.5 (13.8–15.4) | 6.4 (4–9.8) | 20.7 (19.9–21.5) | 20.8 (20–21.7) | 13.1 (9.3–18.1) | 27.9 (27.8–27.9) | 23.6 (23.6–23.7) | 32.9 (32.9–33) |
| Cerebrovascular diseases | 16.6 (16.1–17.1) | 16.9 (16.4–17.5) | 7.3 (5.4–9.6) | 16.2 (15.5–16.9) | 16.5 (15.7–17.3) | 7.3 (4.8–10.9) | 17.1 (16.4–17.8) | 17.4 (16.6–18.2) | 7.1 (4.6–10.7) | 36.3 (36.3–36.4) | 36.5 (36.4–36.5) | 35.4 (35.3–35.5) |
| Influenza & pneumonia | 16.0 (15.5–16.5) | 16.2 (15.7–16.8) | 6.3 (4.6–8.6) | 13.9 (13.3–14.6) | 14.1 (13.4–14.9) | 5.6 (3.4–8.9) | 18.1 (17.4–18.9) | 18.5 (17.7–19.4) | 7.0 (4.5–10.6) | 30.3 (30.3–30.4) | 25.9 (25.9–26) | 36.5 (36.4–36.6) |
| Alzheimer's disease | 11.6 (11.2–12) | 11.7 (11.3–12.2) | 6.5 (4.9–8.6) | 10.3 (9.7–10.9) | 10.3 (9.7–11.1) | 7.5 (44327) | 13.0 (12.3–13.6) | 13.2 (12.5–14) | 5.5 (3.6–8.3) | 44.9 (44.8–45) | 44.7 (44.6–44.7) | 44.4 (44.3–44.5) |
| Chronic liver diseases | 10.4 (10–10.8) | 10.4 (10–10.8) | 6.1 (4.5–8.2) | 7.3 (6.8–7.8) | 7.3 (6.8–7.9) | 4.9 (3.1–7.8) | 13.2 (12.6–13.8) | 13.2 (12.6–14) | 7.3 (4.8–10.9) | 27.5 (27.4–27.5) | 23.3 (23.2–23.3) | 31.9 (31.8–32) |
| Nephritis & nephrosis | 7.2 (6.9–7.6) | 7.3 (7–7.8) | 4.3 (2.8–6.4) | 6.6 (6.1–7.1) | 6.7 (6.2–7.3) | 4.2 (2.3–7.2) | 7.9 (7.4–8.4) | 8.1 (7.6–8.7) | 4.6 (2.4–8) | 16.9 (16.9–17) | 15.2 (15.1–15.2) | 19.4 (19.4–19.5) |
| Chronic lower respiratory diseases | 6.1 (5.7–6.4) | 6.1 (5.8–6.4) | 4.6 (3.1–6.6) | 5.0 (4.7–5.5) | 5.1 (4.7–5.7) | 3.5 (1.8–6.3) | 7.1 (6.6–7.6) | 7.1 (6.6–7.7) | 5.8 (3.5–9.4) | 44.4 (44.4–44.5) | 41.2 (41.1–41.3) | 49.2 (49.1–49.4) |

decreased in NHW males (-252.1 per 100,000 between 2005–2017), NHW females (-125.8 per 100,000), US-born Asian Indian males (-114.2 per 100,000), and US-born Asian Indian females (-52.0 per 100,000). In contrast, all-cause mortality increased in foreign-born Asian Indian males (+38.2 per 100,000) and foreign-born Asian Indian females (+33.8 per 100,000) (**Fig 2**).

## 3.2 Cause-specific mortality

**3.2.1 Females.**   Amongst women, we found significant differences in AMR between foreign-born vs. US-born Asian Indian females in heart disease (AMR 57.0 foreign-born, CI [55.6–58.5] vs. 20.8 US-born, CI[16.3–26.3]), malignant neoplasms (57.0 foreign-born, CI [55.5–58.5] vs. 34.9 US-born, CI[28.8–42.0]), diabetes mellitus (14.5 foreign-born, CI[13.7–15.4] vs. 6.4 US-born, CI[4.0–9.8]), cerebrovascular diseases (17.1 foreign-born, CI[15.7–17.3] vs. 7.6 US-born, CI[4.8–10.9]), and influenza and pneumonia (14.5 foreign-born, CI[13.4–14.9] vs. 7.2 US-born, CI[3.4–8.9]) (**Table 2**) (**Fig 5**).

**3.2.2 Males.**   Amongst males, cause-specific mortality rates were on average 49% higher when compared to corresponding female cause-specific AMR in the aggregated Asian Indian population. We found significant differences in AMR between the foreign-born and US-born Asian Indian male populations in heart disease (AMR 95.8 per 100,000 foreign-born, CI[94.0–

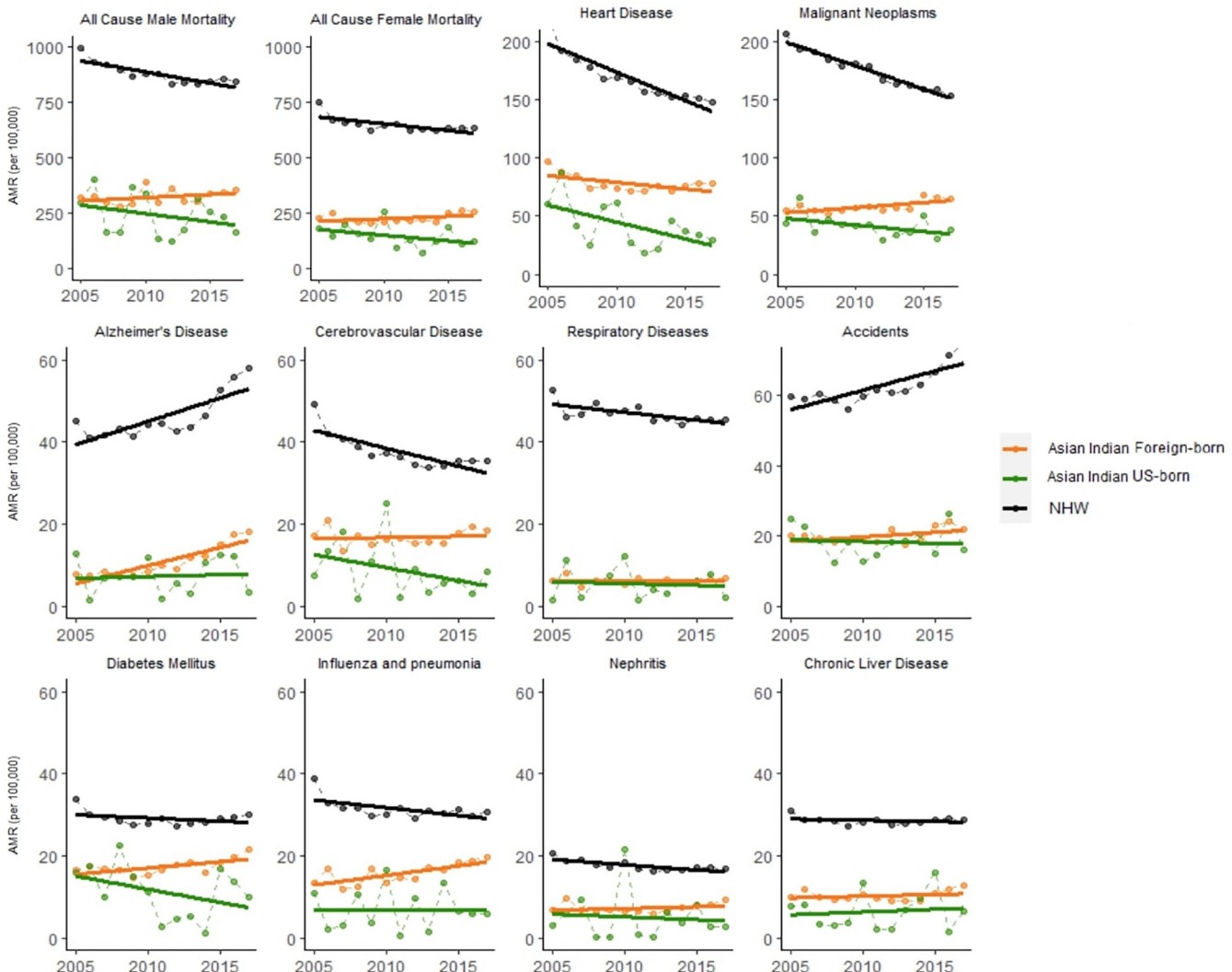

**Fig 1. All-cause and cause-specific age-standardized mortality rates for foreign-born and US-born Asian Indians and non-Hispanic Whites, 2005–2017.**

97.6] vs. 50.2 per 100,000 US-born, CI[42.8–58.7]), malignant neoplasms (62.0 foreign-born, CI[60.6–63.5] vs. 39.8 US-born, CI[33.0–47.8]), diabetes mellitus (20.8 foreign-born, CI[20.0–21.7] vs. 13.1 US-born, CI[9.3–18.0]), cerebrovascular diseases (17.4 foreign-born, CI[16.6–18.2] vs. 7.1 US-born, CI[4.6–10.7]), influenza and pneumonia (18.5 foreign-born, CI[17.7–19.4] vs. 7.0 US-born, CI[4.5–10.6]), Alzheimer's disease (13.2 foreign-born, CI[12.5–14.0] vs. 5.5 US-born, CI[3.6–8.3]), and chronic liver disease (13.2 foreign-born, CI[12.6–14.0] vs. 7.3 US-born, CI[4.8–10.9]) (**Table 2**)(**Fig 5**).

**3.2.3 Age.** The average age of death was over double for foreign-born Asian Indians (70.7 years) compared to US-born (35.1 years), largely explained by only 12.1% of US-born Asian Indians being over the age of 65 at time of death. Average age of death in years for foreign-born Asian Indians was greater in all observed causes including heart disease (74.1 foreign-born, 51.6 US-born), malignant neoplasms (67.3 foreign-born, 43.1 US-born), diabetes mellitus (73.1 foreign-born, 34.7 US-born), cerebrovascular diseases (76.3 foreign-born, 53.1 US-born), influenza and pneumonia (76.7 foreign-born, 31.1 US-born), Alzheimer's disease (77.0

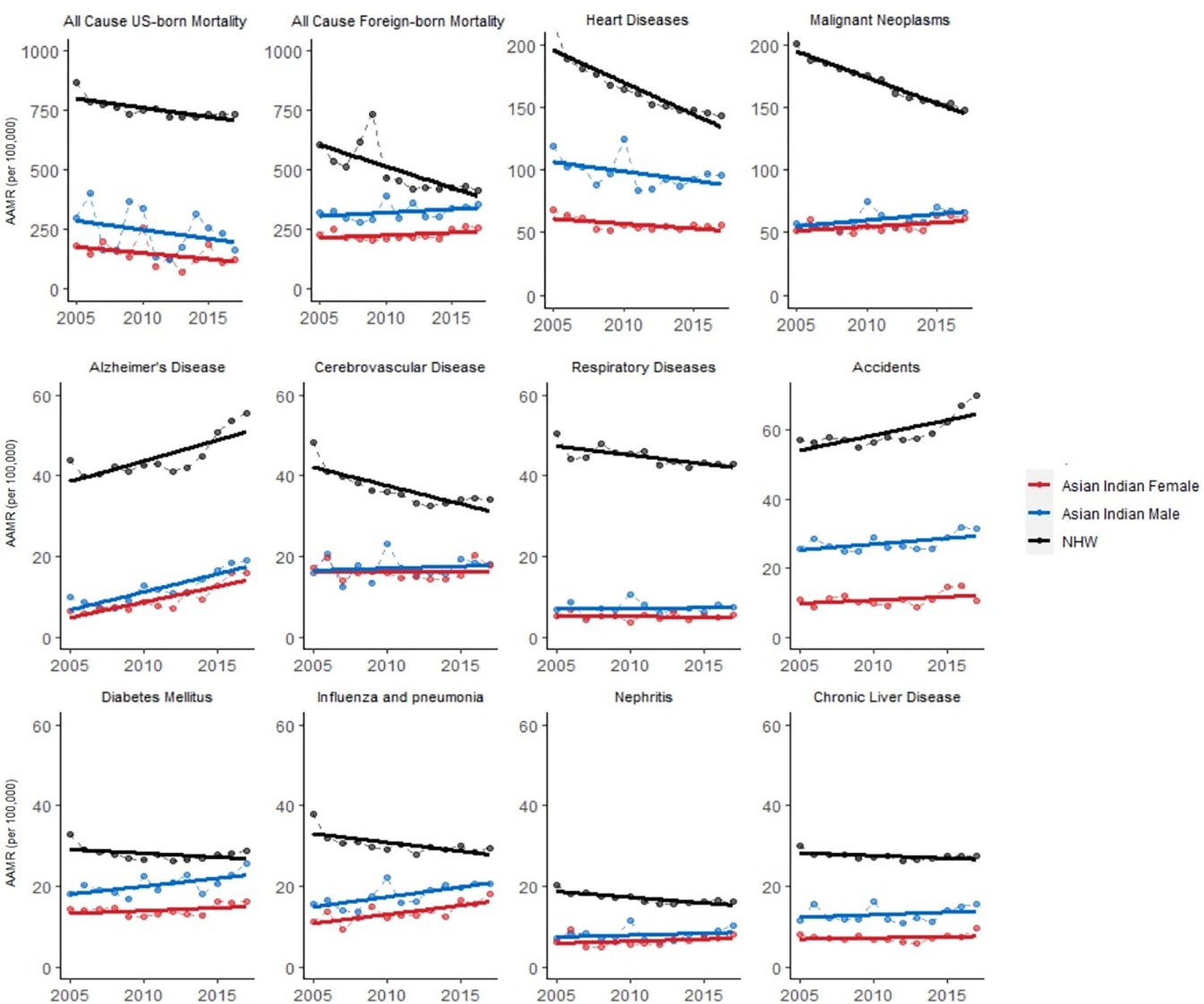

**Fig 2. All-cause and cause-specific age-standardized mortality rate for male and female Asian Indians and non-Hispanic Whites, 2005–2017.**

foreign-born, 27.2 US-born), chronic liver disease (65.4 foreign-born, 31.0 US-born), and accidents (49.7 foreign-born, 23.6 US-born) (Fig 3).

Asian Indian men (66.7 years) also died at a younger average age than Asian Indian women (72.1 years). Average age of death in years for female Asian Indians was greater in all but one of the observed causes, including heart disease (70.8 male, 78.3 female), diabetes mellitus (69.4 male, 75.2 female), cerebrovascular diseases (73.11 male, 78.7 female), influenza and pneumonia (74.7 male, 75.5 female), Alzheimer's disease (72.7 male, 74.9 female), chronic liver disease (60.8 male, 69.7 female), and accidents (43.7 male, 49.5 female). Asian Indian women died of malignant neoplasms (67.4 male, 65.4 female) at a younger age (Fig 4).

**3.2.4 Foreign-born vs. US-born.** Examining cause-specific mortality rates, AMR was higher in the foreign-born Asian Indian population compared to the US-born for all causes of death (Fig 1) (Fig 5): heart disease (AMR 76.9 per 100,000 foreign-born, CI[75.8–78.1] vs. 35.2 per 100,000 in US-born, CI[30.9–40.1]), malignant neoplasms (59.3 foreign-born, CI[58.3–

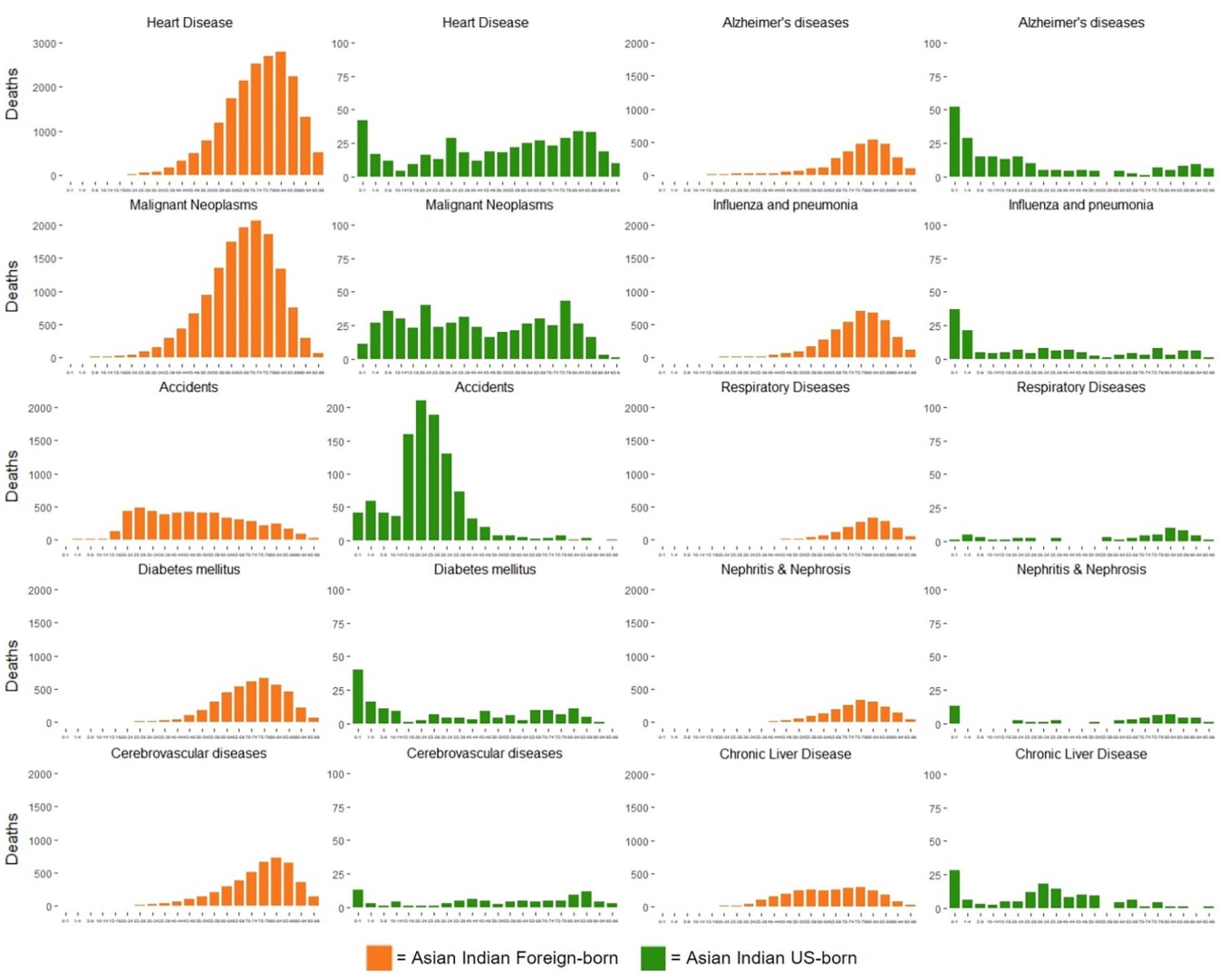

**Fig 3. Age distribution for leading causes of death in foreign-born and US-born Asian Indians, 2005–2017.**

60.3] vs. 37.1 US-born, CI[32.5–42.3]), accidents (20.2 foreign-born, CI[19.5–20.9] vs. 17.8 US-born, CI[15.7–20.4]), diabetes mellitus (17.7 foreign-born, CI[17.2–18.3] vs. 9.7 US-born, CI[7.4–12.5]), cerebrovascular disease (16.9 foreign-born, CI[16.4–17.5] vs. 7.3 US-born, CI [5.4–9.6]), influenza and pneumonia (16.2 foreign-born, CI[15.7–16.8] vs. 6.3 US-born, CI [4.6–8.6]), Alzheimer's disease (11.7 foreign-born, CI[11.3–12.2] vs. 6.5 US-born, CI[4.9–8.6]), chronic liver diseases (10.4 foreign-born, CI[10.0–10.8] vs. 6.1 US-born, CI[4.5–8.2]), and nephritis and nephrosis (7.3 foreign-born, CI[7.0–7.8] vs. 4.3 US-born, CI[2.8–6.4]). (**Table 2**).

## 4. Discussion

Between 2005 and 2017, heart disease (AMR 75.6 per 100,000, CI[74.5–76.7]) was the leading cause of death amongst Asian Indians in the United States, followed by malignant neoplasms (AMR 58.6, CI[57.7–59.6]) and accidents (AMR 19.6, CI[19.1–20.2]), which accounted for 25.4%, 22.0% and 8.5% of deaths in Asian Indians, respectively. Mortality rates from malignant

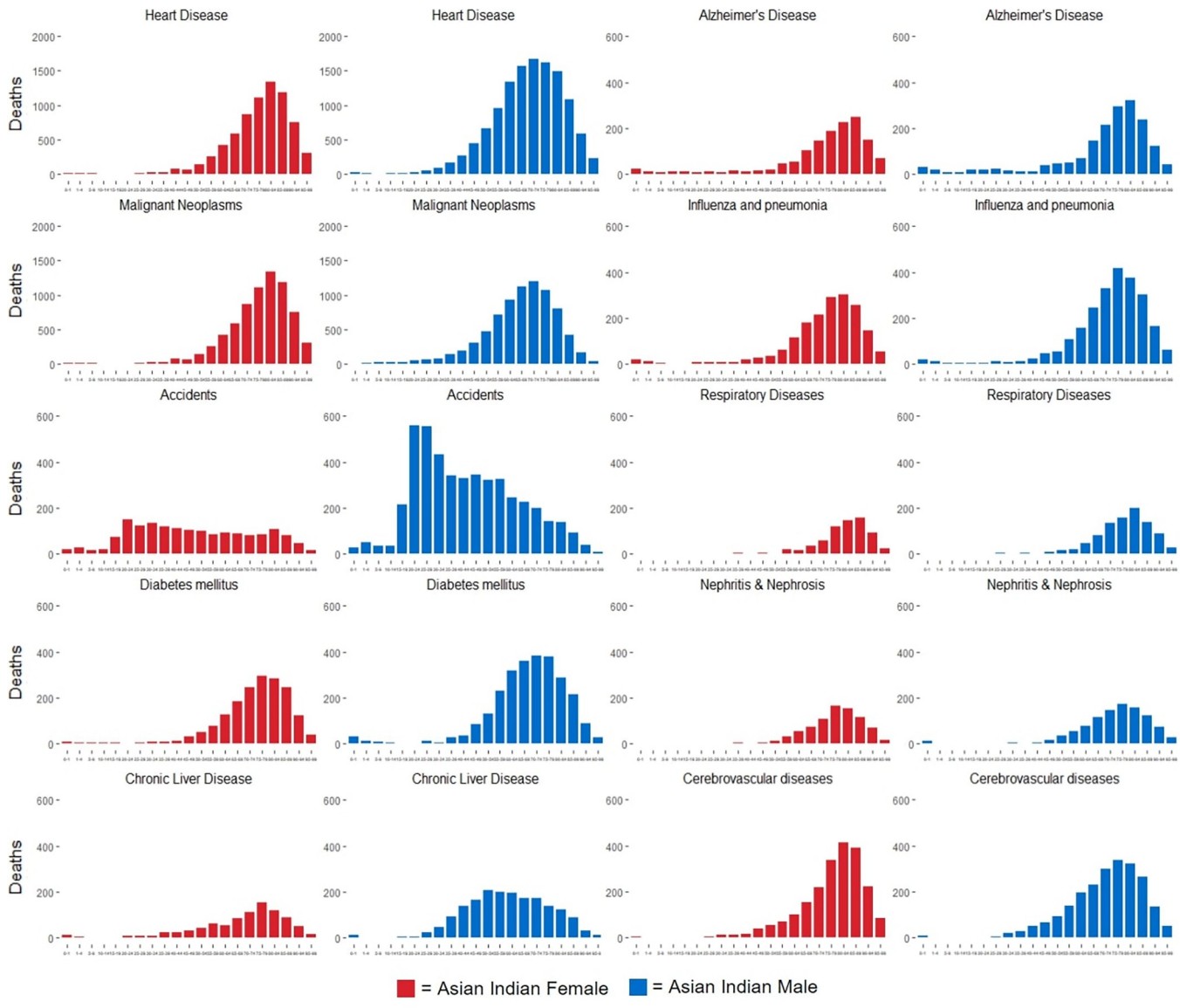

**Fig 4. Age distribution for leading causes of death in male and female Asian Indians, 2005–2017.**

neoplasms, influenza/pneumonia, and chronic liver disease increased for Asian Indians during this time period compared to NHWs, in whom these mortality rates decreased. Alzheimer's disease mortality rate increased by 129% for foreign-born Asian Indians over this time period, compared to a 27% increase in NHWs, perhaps related to the aging foreign-born Asian Indian population. All-cause mortality rates decreased for both aggregated Asian Indians, US-born Asian Indians, and NHWs, but increased for foreign-born Asian Indians.

Differences in leading causes of death were observed by sex and place of birth. Malignant neoplasms (AMR 56.2 per 100,000) was the leading cause of death in Asian Indian females, while heart disease (AMR 94.5 per 100,000) was the leading cause in Asian Indian males. For foreign-born Asian Indian females, heart disease and malignant neoplasms were tied as the leading causes of death (AMR 57.0 per 100,000), while for US-born Asian Indian females, malignant neoplasms was the sole leading cause of death (AMR 34.9 per 100,000). In Asian

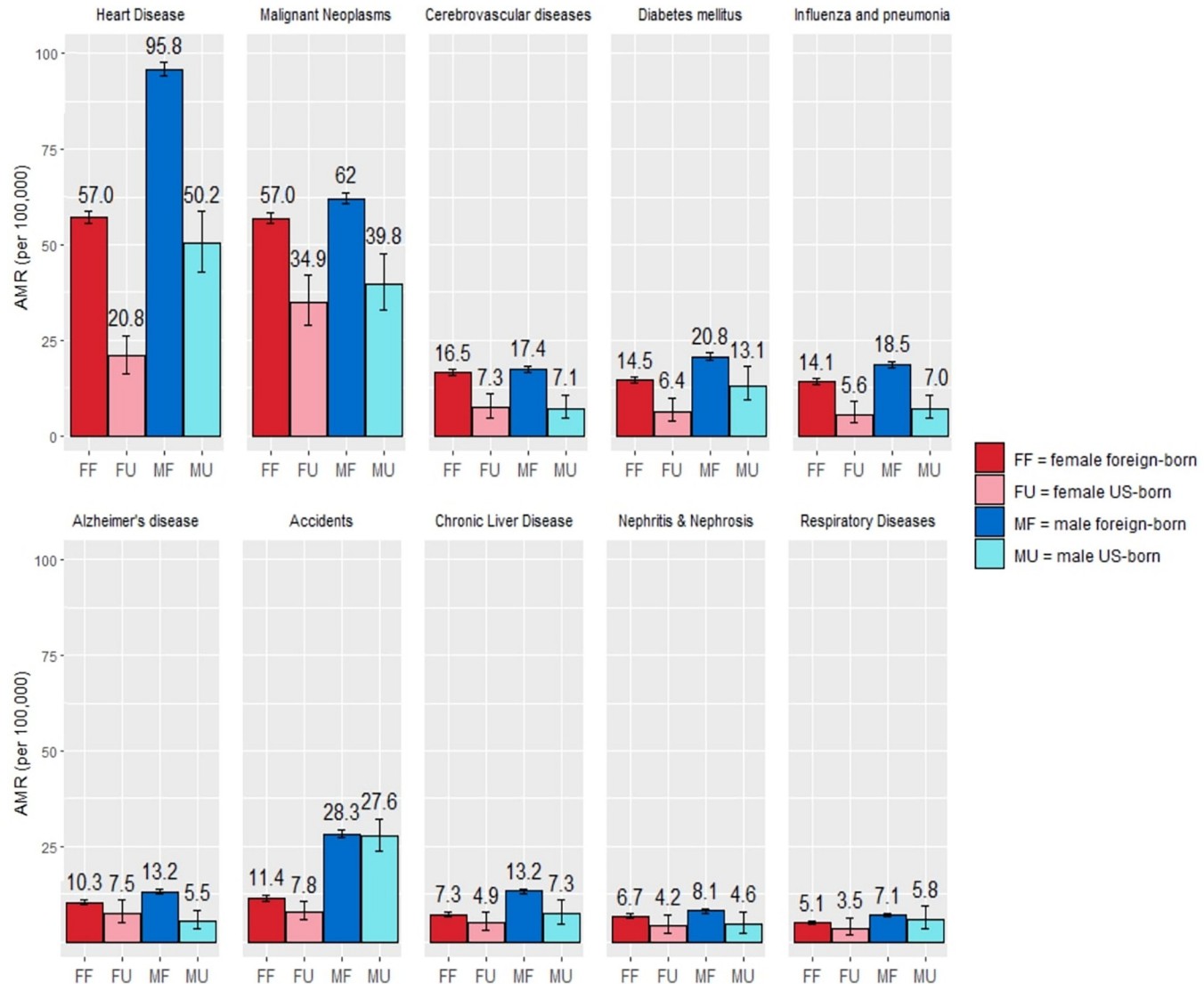

**Fig 5. Cause-specific age-standardized mortality rates (AMR) in Asian Indians by place of birth for in the United States, 2005–2017.**

Indian males, heart disease was the sole leading cause of death (AMR 95.8 per 100,000 in foreign-born, 50.2 per 100,000 in US-born). For foreign-born Asian Indians, heart disease was the leading cause of death (AMR 76.9 per 100,000), while for US-born Asian Indians, the leading cause of death was malignant neoplasms (AMR 37.2 per 100,000). Additionally, foreign-born Asian Indians died at a higher rate from all causes than US-born Asian Indians throughout this period, but they died at an older average age (70.7 foreign-born, 35.0 US-born). The significantly lower average age of death for US-born Asian Indians arises from the younger age demographics of this group, explained by historical and migratory trends [18, 19].

Heart disease is the leading cause of death in South Asians because of higher burden of cardiovascular risk factors compared to NHWs [25], which are related to cultural differences in dietary patterns, physical activity, genetics, and response to medications [5, 11, 26]. For instance, the Mediators of Atherosclerosis in South Asians Living in America (MASALA) study [27] examined cardiovascular health factors and behaviors in middle-aged South Asians

living in America (predominantly foreign-born Asian Indians). In this study, only 11% of those aged 40–59 years old achieved ideal levels in five or more out of the seven cardiovascular health (CVH) metrics, such as smoking and diet [28], with only 2.4% of the MASALA participants having dietary quality in the ideal range. However, since adults aged 65 or older are more likely to develop heart disease compared to younger adults [29], the high rate of heart disease mortality in foreign-born Asian Indians observed is likely in part due to the older age distribution. Nonetheless, prior research has shown that Asian Indians also have a high burden of premature mortality from heart disease, evidenced by high years of potential life lost due to ischemic heart disease in Asian Indians [30]. Heart disease mortality rates decreased between 2005–2017 in both foreign-born and US-born Asian Indians, which is attributable to clinical and public health improvements in primary and secondary heart disease prevention. However, we still see a more notable decrease in heart disease mortality during this period for US-born Asian Indians when compared to foreign-born, suggesting an important contribution of cultural and socioeconomic factors towards heart disease mortality risk. Developing interventions that address the cultural and socioeconomic determinants of health and modifiable risk factors that are contributing to this observed trend is critical to implement clinical and public health practices that can improve the health and wellbeing of specific ethnic subgroups.

The mortality rate from malignant neoplasms was relatively stable over this time period, slightly increasing in the foreign-born (+18%) and slightly decreasing in the US-born (-12%) Asian Indian population. However, there is large heterogeneity between the age distributions of the US-born Asian Indian population compared to the foreign-born (Fig 3), which determines the types of cancers contributing to the mortality rate in each population and which consequently influence trends, since different types of cancer have differing mortality rates. Other factors such as screening rates and accessibility to preventive and other health services might have also played a role in the observed trends. Foreign-born Asian Indians are likely to face more barriers in accessing health care services, which leads to worse health outcomes. It is imperative that the increase in mortality rate from malignant neoplasms among the foreign-born Asian Indian subgroup be further explored and that appropriate clinical and public health interventions addressing the underlying risk factors be developed accordingly. Additionally, Asian Indian accident-related mortality has distinctly higher mortality rates in the younger age brackets, but overall has increased in the foreign-born population (+10.1%) and decreased in the US-born population (-6.2%) over this time period. Further research is needed to elucidate these trends.

Unlike the "healthy immigrant" effect seen in the Hispanic/Latino population in the US [31], in which foreign-born Hispanic/Latinos seem to have better health than their US-born counterparts, foreign-born Asian Indians have higher mortality rates than US-born Asian Indians. Multiple explanations have been posited for the "healthy immigrant" paradox, for instance individuals' returning to their home countries when ill/approaching death, which biases mortality estimates. It is unclear the extent to which similar factors operate in the Asian Indian population in the US impacting mortality rates in foreign-born Asian Indians. The younger age distribution in the US-born Asian Indian population likely plays a role in the differences by place of birth, as the average age of death in years was over double for foreign-born (70.7) compared to US-born (35.0) Asian Indians, and we observe greater mortality rates in older populations. Generational differences in health-related behaviors may also explain a portion of the foreign-born/US-born mortality gap. Heart disease and malignant neoplasms account for a majority of the difference in mortality between the foreign-born and US-born Asian Indian populations, with heart disease alone accounting for 38% of this difference, and heart disease and malignant neoplasms combined accounting for 59% of the difference. This is important, as these two conditions affect older people more often, posing a greater risk for the demographically older foreign-born Asian Indian population. Collectively, the differences in

mortality patterns may reflect cultural differences related to behavior such as diet and physical activity, including higher carbohydrate- and fat-content diets and a more sedentary lifestyle [32], which influences both heart disease and cancer-related mortality [28], and differences in social determinants of health between foreign-born and US-born Asian Indians. Although a relatively small proportion of the Asian Indian population is uninsured (approximately 6% after the Affordable Care Act in 2015) [33, 34], health care access differences related to health literacy, language accessibility, and utilization of healthcare services may also influence differences in mortality between the foreign-born and US-born Asian Indian populations. There are also notable disparities in cancer screening among Asian Indians, with this population facing several sociocultural barriers to access of this type of care, including individual and structural barriers as well. Further research would be needed to understand how this translates into differences in cancer-related mortality between foreign-born and US-born Asian Indians [35]. These observations are important for both clinical and public health practices, since interventions will need to account for these different trends, demographic factors, and barriers to effectively target the subgroup-specific problems affecting foreign-born and US-born Asian Indians differently, as well as the underlying risk factors involved. These findings can offer insights and guide further research to expand understanding of all these determinants and to develop culturally sensitive practices that reduce these disparities [35].

This study has several strengths and limitations. The National Vital Statistics program [22] captures all deaths in the United States and contains disaggregated Asian ethnicity data to understand mortality patterns in Asian American subgroups. However, causes of death, race/ethnicity and other key fields on death certificates may be unintentionally misclassified by funeral directors, coroners or reporting physicians, the individuals responsible for their completion [7, 36]. This information is supplied by next-of-kin, but if unavailable, this field is otherwise completed by the funeral directors, coroners or reporting physicians. This method may lead to incorrect racial/ethnic categorization, potentially leading to other South Asian nationalities (including Pakistani, Nepalese, and Bhutanese) being classified as Asian Indian [37], among other potential critical errors. Moreover, multiracial individuals might be assigned to a single racial group [38], further obscuring trends in ethnic subgroups. Improving national surveillance systems to accurately represent these ethnic subgroups is also essential for accurate characterization of health trends and improved clinical and public health practices targeting these subgroups of the population.

## 5. Conclusion

Leading causes of death in Asian Indians in the US between 2005–2017 were heart diseases, malignant neoplasms, and accidents or unintentional injuries, with differences noted by sex and place of birth. Further work to identify the biological and sociocultural factors related to immigration and acculturation is warranted to better understand differences in mortality related to nativity in this population. Additional research into mortality rates of Asian Indians using the 2020 census data for the population is also important. Ultimately, the results of this study begin to inform the critical need for culturally appropriate prevention initiatives and clinical practices, as well as public policy development targeted towards the leading causes of mortality in individual ethnic subgroups in order to support health equity and ensure adequate resource allocation and health spending.

## Supporting information

**S1 Table. Annual mortality ratio for leading causes of death in Asian Indians and non-Hispanic Whites in the United States by nativity, 2005–2017.**
(DOCX)

**S2 Table. Annual mortality ratio for leading causes of death in Asian Indians and non-Hispanic Whites in the United States by gender, 2005–2017.**
(DOCX)

## Author Contributions

**Conceptualization:** Claudia Fernandez Perez, Kevin Xi, Aditya Simha, Nilay S. Shah, Robert J. Huang, Latha Palaniappan, Nora Sharp, Malathi Srinivasan.

**Data curation:** Kevin Xi, Tim Au, Nora Sharp, Nathaniel Islas.

**Formal analysis:** Kevin Xi, Sukyung Chung, Nathaniel Islas.

**Funding acquisition:** Latha Palaniappan, Malathi Srinivasan.

**Investigation:** Tim Au, Nora Sharp.

**Methodology:** Kevin Xi, Nilay S. Shah, Sukyung Chung, Nora Sharp, Nathaniel Islas, Malathi Srinivasan.

**Project administration:** Latha Palaniappan, Malathi Srinivasan.

**Software:** Kevin Xi.

**Supervision:** Nilay S. Shah, Robert J. Huang, Latha Palaniappan, Sukyung Chung, Malathi Srinivasan.

**Validation:** Sukyung Chung.

**Visualization:** Claudia Fernandez Perez, Kevin Xi, Aditya Simha.

**Writing – original draft:** Claudia Fernandez Perez, Kevin Xi, Aditya Simha.

**Writing – review & editing:** Claudia Fernandez Perez, Kevin Xi, Aditya Simha, Nilay S. Shah, Robert J. Huang, Latha Palaniappan, Sukyung Chung, Tim Au, Nora Sharp, Nathaniel Islas, Malathi Srinivasan.

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
