## [Decision Letter · Decision Letter 0]

22 Sep 2021

PONE-D-21-08873Leading Causes of Death in Asian Indians 2005-2017: Mortality rates across nativity, age, and sexPLOS ONE

Dear Dr. Fernandez Perez,

Thank you for submitting your manuscript to PLOS ONE. After careful consideration, we feel that it has merit but does not fully meet PLOS ONE’s publication criteria as it currently stands. Therefore, we invite you to submit a revised version of the manuscript that addresses the points raised during the review process.

We look forward to receiving your revised manuscript.

Kind regards,

Jianhong Zhou

Associate Editor

PLOS ONE

Journal Requirements:

2.Please provide additional details regarding participant consent.

5 Please amend either the title on the online submission form (via Edit Submission) or the title in the manuscript so that they are identical.

6. Please ensure that you refer to Figure 3  in your text as, if accepted, production will need this reference to link the reader to the figure.

Reviewers' comments:

Reviewer's Responses to Questions

**Comments to the Author**

1. Is the manuscript technically sound, and do the data support the conclusions?

Reviewer #1: Yes

Reviewer #2: Yes

2. Has the statistical analysis been performed appropriately and rigorously? 

Reviewer #1: Yes

Reviewer #2: Yes

3. Have the authors made all data underlying the findings in their manuscript fully available?

Reviewer #1: No

Reviewer #2: Yes

4. Is the manuscript presented in an intelligible fashion and written in standard English?

Reviewer #1: Yes

Reviewer #2: Yes

5. Review Comments to the Author

Reviewer #1: The data appear to be HIPAA restricted data but it says authors with appropriate credentials may access the data. It was a little difficult to understand the age of mortality item in the earlier section of the paper though it is explained in the discussion. Maybe where it is first introduced, the authors might refer the reader to the later discussion or give a brief explanation? Additionally, now that the 2020 Census data are coming out, it could be good to note that and state additional examination of mortality for this key population should continue for the data to come using the 2020 population denominators etc. I understand use of ACS data for the later years in the current data set.

Reviewer #2: Overall, this is a well-conducted study characterizing patterns of mortality among an understudied yet at-risk population group in the United States. This builds upon prior longitudinal data investigating trends among Asian Indians to identify particularly vulnerable populations. My review largely surrounds the implications of this work for clinical and public health practice, along with minor comments, as enumerated below:

*The authors should provide a citation about the drivers of immigration for Asian Indians, both in the 1960s and more recently. The claims are made without references.

*The authors assert that blending of Asian Indian and American cultures may impact risks, but don't expand upon why specifically for this population group. In addition, they also state that this consideration supports "place of birth" as an important health determinant, which implies the presumption that this is only or primarily true for immigrants.

*Outside of being "in accordance with survey data", the rationale for special age brackets is unclear. Please provide more context.

*The US-born Asian Indian average age of death seems shocking low (35.1) and it is unclear how Figure 4 makes this clear (the title of the figure reflects age distributions by sex for selected endpoints). I would urge the authors to make sure that value is accurate. If so, this requires some attention in the Discussion.

*Building off the prior comment, I find the Discussion to be primarily a repeat of the results. Although it was not measured in this study, I would like to see the results placed in more nuanced context of the extant literature. Especially with respect to modifiable risk factors that have socio-cultural underpinnings, these need to be described more fully (currently there are generic statements about diet, physical activity, genetics, and response to medication). If the reader is to be a health professional, what do these results mean for intervention and prevention?

*Similarly, more detail about differential distributions of cancer among Asian Indians would be beneficial, as this study treats cancer death as a singular outcome.

*The idea that this study is in contrast with a "healthy immigrant" effect is compelling, and I wish more attention was provided to elucidate this; currently, the authors make general comments about generational and cultural differences related to behavior, along with health care access differences. I think an opportunity is lost for this paper to provide concrete directions for clinical and public health practice, as opposed to a reiteration of the findings in the final sections of the manuscript.

6. PLOS authors have the option to publish the peer review history of their article (what does this mean?). If published, this will include your full peer review and any attached files.

Reviewer #1: No

Reviewer #2: No

---

## [Author Response · Author response to Decision Letter 0]

24 Jan 2022

All points addressed in "Response to Revisions" document uploaded as additional Cover Letter.

---

## [Editor Report · Decision Letter 1]

28 Feb 2022

PONE-D-21-08873R1Leading Causes of Death in Asian Indians in the United States (2005 - 2017)PLOS ONE

Dear Dr. Fernandez Perez,

Thank you for submitting your manuscript to PLOS ONE. After careful consideration, we feel that it has merit but does not fully meet PLOS ONE’s publication criteria as it currently stands. Therefore, we invite you to submit a revised version of the manuscript that addresses the points raised during the review process. First off my apologies for the delay.  Your original handling editor needed to hand off this assignment and I stepped in to take over earlier this week.  I am sending this back to you to give you a chance to revise your rebuttal letter.  The one that your submitted makes no reference to any of the issues raised by reviewer two, instead mentioning only some minor revisions in response to editorial issues.  I have reproduced the original comments of reviewer two below for easy reference.  Please detail what revisions were made in response to each of these criticisms.  Please also keep in mind that reviewer two will be invited to review your revision. 

We look forward to receiving your revised manuscript.

Kind regards,

Patrick R Stephens, Ph.D.

Academic Editor

PLOS ONE

Comments from reviewer two:

Overall, this is a well-conducted study characterizing patterns of mortality among an understudied yet at-risk population group in the United States. This builds upon prior longitudinal data investigating trends among Asian Indians to identify particularly vulnerable populations. My review largely surrounds the implications of this work for clinical and public health practice, along with minor comments, as enumerated below:

*The authors should provide a citation about the drivers of immigration for Asian Indians, both in the 1960s and more recently. The claims are made without references.

*The authors assert that blending of Asian Indian and American cultures may impact risks, but don't expand upon why specifically for this population group. In addition, they also state that this consideration supports "place of birth" as an important health determinant, which implies the presumption that this is only or primarily true for immigrants.

*Outside of being "in accordance with survey data", the rationale for special age brackets is unclear. Please provide more context.

*The US-born Asian Indian average age of death seems shocking low (35.1) and it is unclear how Figure 4 makes this clear (the title of the figure reflects age distributions by sex for selected endpoints). I would urge the authors to make sure that value is accurate. If so, this requires some attention in the Discussion.

*Building off the prior comment, I find the Discussion to be primarily a repeat of the results. Although it was not measured in this study, I would like to see the results placed in more nuanced context of the extant literature. Especially with respect to modifiable risk factors that have socio-cultural underpinnings, these need to be described more fully (currently there are generic statements about diet, physical activity, genetics, and response to medication). If the reader is to be a health professional, what do these results mean for intervention and prevention?

*Similarly, more detail about differential distributions of cancer among Asian Indians would be beneficial, as this study treats cancer death as a singular outcome.

*The idea that this study is in contrast with a "healthy immigrant" effect is compelling, and I wish more attention was provided to elucidate this; currently, the authors make general comments about generational and cultural differences related to behavior, along with health care access differences. I think an opportunity is lost for this paper to provide concrete directions for clinical and public health practice, as opposed to a reiteration of the findings in the final sections of the manuscript.
---

## [Author Response · Author response to Decision Letter 1]

31 May 2022

Initial response to reviewers letter is included as "INITIAL Response to Reviewers 1" and latest response to comments letter is included as "Response to Revisions 2" or similar names, but with versions clearly indicated. Please reach out if further clarification of earlier stages of the revision process is needed, or with any other additional comments/inquiries.

---

## [Decision Letter · Decision Letter 2]

14 Jun 2022

PONE-D-21-08873R2

Leading Causes of Death in Asian Indians in the United States (2005 - 2017)

PLOS ONE

Dear Dr. Fernandez Perez,

Thank you for submitting your manuscript to PLOS ONE. After careful consideration, we feel that it has merit but does not fully meet PLOS ONE’s publication criteria as it currently stands. Therefore, we invite you to submit a revised version of the manuscript that addresses the points raised during the review process.

First off, my sincere apologies with the confusion around revisions to the previous versions and the delays that resulted from this.  The reviewer that asked for major revisions has had a chance to look at the updated manuscript now, and is happy with your updates.  The reviewer has also suggested a minor update to the discussion to make the recommendations to workers in the field more clear, particularly for HCWs that service Asian Indian populations.  I am sending this back to you one more time to give you a chance to implement this suggestion.  However, I leave the extent of the revision to your discretion since the manuscript is already technically sound.  I do not anticipate needing to send your final revision out for further review.

We look forward to receiving your revised manuscript.

Kind regards,

Patrick R Stephens, Ph.D.

Academic Editor

PLOS ONE

Journal Requirements:

Reviewers' comments:

Reviewer's Responses to Questions

**Comments to the Author**

1. If the authors have adequately addressed your comments raised in a previous round of review and you feel that this manuscript is now acceptable for publication, you may indicate that here to bypass the “Comments to the Author” section, enter your conflict of interest statement in the “Confidential to Editor” section, and submit your "Accept" recommendation.

Reviewer #2: All comments have been addressed

2. Is the manuscript technically sound, and do the data support the conclusions?

Reviewer #2: Yes

3. Has the statistical analysis been performed appropriately and rigorously? 

Reviewer #2: Yes

4. Have the authors made all data underlying the findings in their manuscript fully available?

Reviewer #2: Yes

5. Is the manuscript presented in an intelligible fashion and written in standard English?

Reviewer #2: Yes

6. Review Comments to the Author

Reviewer #2: For the most part, the concerns I articulated in my initial review have been addressed. However, in the Discussion, I would have liked to see more specificity in the recommendations for public health practice. For instance, there is a wealth of literature looking at specific dietary patterns among Asian Indians that could have been emphasized, or social/cultural barriers to cancer screening. Currently, the implications for disease prevention and treatment still remain in the realm of generic recommendations which could be applied to any population group.

I defer to the editors to determine if and how this sections could be more concrete and targeted, given the population-specificity of this work.

I look forward to seeing this manuscript as a publication in the base of peer-reviewed literature in the near future.

7. PLOS authors have the option to publish the peer review history of their article (what does this mean?). If published, this will include your full peer review and any attached files.

Reviewer #2: No

---

## [Author Response · Author response to Decision Letter 2]

28 Jun 2022

Thank for suggesting this review to strengthen the discussion section and recommendations to healthcare practitioners. Additional information has been incorporated to reflect the notable cancer disparities among Asian Indians, as well as the sociocultural barriers that they face. It’s also been pointed out that there is crucial need for further research into how this translates into differences in cancer related mortality between foreign-born and US-born Asian Indians. We had also incorporated information about specific diet related and other lifestyle factors that can be at play in the broad disparities that we see among this population. We further note how these findings can guide further research and practitioners in developing culturally sensitive practices that reduce these disparities.

---

## [Editor Report · Decision Letter 3]

30 Jun 2022

Leading Causes of Death in Asian Indians in the United States (2005 - 2017)

PONE-D-21-08873R3

Dear Dr. Fernandez Perez,

We’re pleased to inform you that your manuscript has been judged scientifically suitable for publication and will be formally accepted for publication once it meets all outstanding technical requirements.  Congratulations, and my apologies again that this took so long.

Kind regards,

Patrick R Stephens, Ph.D.

Academic Editor

PLOS ONE

---

## [Editor Report · Acceptance letter]

14 Jul 2022

PONE-D-21-08873R3 

Leading Causes of Death in Asian Indians in the United States (2005 - 2017) 

Dear Dr. Fernandez Perez:

I'm pleased to inform you that your manuscript has been deemed suitable for publication in PLOS ONE. Congratulations! Your manuscript is now with our production department. 

Kind regards, 

on behalf of

Dr. Patrick R Stephens 

Academic Editor

PLOS ONE